# Effectiveness of four antiretroviral regimens for treating people living with HIV

**Aracele Tenório de Almeida e Cavalcanti**[1,2]*, **Ricardo Arraes de Alencar Ximenes**[2,3], **Ulysses Ramos Montarroyos**[2], **Polyana Monteiro d'Albuquerque**[4,5], **Rosário Antunes Fonseca**[6], **Demócrito de Barros Miranda-Filho**[2,4]

**1** Clinical Hospital, Federal University of Pernambuco, Recife, Brazil, **2** Postgraduate Program in Health Sciences, University of Pernambuco, Recife, Brazil, **3** Tropical Medicine, Federal University of Pernambuco, Recife, Brazil, **4** Internal Medicine, University of Pernambuco, Recife, Brazil, **5** Correia Picanço Hospital, Recife, Brazil, **6** Nossa Senhora das Graças Faculty of Nursing, University of Pernambuco, Recife, Brazil

* aracelecavalcanti@hotmail.com

**Data Availability Statement:** In accordance with the standards for good clinical research of the HUOC/Procape Hospital Complex Ethics

## Abstract

The aim of this study was to compare 4 different ARV regimens in a clinical cohort in Brazil, with regard to the virologic and immunologic responses, clinical failure and reasons for changing. To compare the virologic response and clinical failure between groups we used the Cox and Kaplan Meier proportional hazard models. To analyze the immunologic outcome, we used multilevel GLLAMM and mixed effect linear regression models. To compare regimen change outcomes we used the Pearson's chi-square test. We included 840 participants distributed across the groups according to the initial ART regimen. The mean follow-up period was 27.8 months. Almost half the sample initiated ART with AIDS-related signs/ symptoms. Virologic response was effective in 79.6% of participants within 12 months. The tenofovir/lamivudine/efavirenz group presented a higher proportion of virologic response (VL<50 at 6 months) when compared to the zidovudine/lamivudine/efavirenz group. There was no difference between the regimens regarding the immunologic response. A total of 17.3% of individuals changed regimen because of failure and 46.5% due to adverse events. Changes due to adverse events were more frequent in the group using zidovudine/lamivudine/efavirenz. The proportion of hospitalizations at 1 year was higher in the zidovudine/lamivudine/efavirenz group when compared to the tenofovir/lamivudine/efavirenz group. The effectiveness outcomes between the regimens were similar. Some differences may be due to the individual characteristics of patients, toxicity and acceptability of drugs. Studies are needed that compare similarly effective regimens and their respective treatment costs and financial impacts on SUS (Integrated Healthcare System).

## Introduction

Antiretroviral therapy (ART) has changed the prognosis of AIDS, and has increased longevity and decreased mortality and morbidity [1–3]. The decision regarding the choice of antiretroviral regimen should consider factors such as: efficacy, immediate and long-term toxicity; the

Committee, data cannot be shared publicly because the dataset contains sensitive human subject data. Participants did not provide consent for public sharing of their data, and public availability would compromise patient privacy. De-identified data can made available upon reasonable request from qualified investigators by contacting email address ppg.cienciasdasaude@upe.br.

**Funding:** Funding: Ministry of Health, Secretary of Health Surveillance (SVS) -Edital No. 20/2013, Agreement 796577/2013, Applied Studies and Research in Health Surveillance. This study was funded in part by the Higher Education Personnel Improvement Coordination - Brazil (CAPES) - Finance Code 001. National Council for Scientific and Technological Development (309722 / 2017-9 to RAAX, 308590 / 2013-9 to DBMF)

**Competing interests:** The authors have declared that no competing interests exist.

presence of co-infections and comorbidities; the concomitant use of other medicines; potential of adherence; adequacy to the patient's everyday routine; genetic barrier to drug resistance; medication and food interactions and the cost of medication [2, 4]. The effectiveness of the regimens is most often measured by their ability to suppress the viral load and restore immunity, although this may be affected by toxicity and adverse events, which often leads to a modification of treatment [5, 6]. The reasons for virologic failure are complex, although poor adherence to medication is the most common cause [7].

From 2013, based on the perspective of reducing HIV transmissibility, Brazil was the third country worldwide to encourage the initiation of ART regardless of the T-CD4 lymphocyte count, considering the motivation of the patient [8]. Thus, treatment protocols for people living with HIV (PLWHIV) in Brazil have advanced in relation to adopting regimens with a lower potential for adverse events, toxicity, a higher genetic barrier to resistance and dosage convenience. Therefore, in countries where there is universal access to ART, it is of great importance to monitor the effectiveness of different regimens frequently recomended in the guidelines, taking into account individual population characteristics.

In this study, we evaluated the effectiveness of different initial antiretroviral regimens, comparing groups for virologic response, immunologic response, clinical failure, and reasons for regimen changes in a cohort with PLWHIV, monitored at two outpatient services in Recife, Brazil.

## Material and methods

This was a bidirectional clinical cohort followed at the Hospital Universitário Oswaldo Cruz and Hospital Correia Picanço, in Recife/PE, Brazil. Together, the two hospitals take in around half the adult HIV/AIDS cases in the state of Pernambuco, in outpatient follow-up and also provide beds for hospitalization, including intensive care.

### Sample planning and ethical considerations

The patient universe registered at SICLOM (the drug logistic management system of the Brazilian Ministry of Health) from the start of collection (January 5, 2015) made up the prospective phase of the study. Of the 626 registrations, 519 patients were eligible for the study. The sample for the retrospective component was obtained from a simple random draw of the records in the logistic drug control system of the two health services for a number of participants proportional to that of the prospective component. The filter for eligibility of registrations was to be over 18 years old and to start antiretroviral therapy with any of the four regimens that would be studied. The number "six" was randomly drawn and every six number records, one was selected to compose the sample. Of the 2196 eligible participants, 472 were drawn (Fig 1). We included PLHIV over 18 years old who started their first ART regimen between January 2, 2011 and April 30, 2016. Follow-up information was obtained through May 30, 2017. Participants were guaranteed complete anonymity at the time of the draw, where they were selected through the system registration number and medical record number. All selected participants were identified by system registration numbers and their identity preserved. The study was approved by the HUOC / Procape Hospital Complex Ethics Committee [CAAE No. 30658514.9.0000.5192—Opinion Number: 697.040] and complied with the requirements of Resolution No. 466/2012 of the National Health Council, which establishes standards for research involving human beings, with exemption from the Informed Consent Form (ICF).

The ART regimens compared were: tenofovir/lamivudine/efavirenz; zidovudine/lamivudine/efavirenz; tenofovir/lamivudine/atazanavir/ritonavir and tenofovir/lamivudine/lopinavir/ritonavir. We considered losses participants whose medical records did not allow follow-

## Sample Selection Algorithm

**Fig 1. Sample selection.**

up (some patients are registered at the system but are followed up in private services; others may have been transferred to another health service). Women who changed ART in the first year due pregnancy were accounted in the "losses", and followed up at other health services. The total loss was 151 patients, thereby resulting in a total of 840 participants.

### Procedures for data collection

Participants were monitored from the moment that ART had been introduced, through a review of outpatient records and from access systems for drug dispensing and examinations. Data for the CD4 and viral load counts were retrieved from medical records and databases from SISCEL (the laboratory examination control system). In order to investigate genotyping and access the confirmation reports for cases of resistance mutations, in addition to the medical records, we also consulted SISGENO (the system and information for the genotyping network). Information on hospitalization and death from any cause was taken from the Hospitalization Information System (SIH) and from the Mortality Information System (SIM), both from the Brazilian Ministry of Health. Between Jan/2011 and May/2017, we searched the SIH database to identify individuals, date of hospitalization, diagnostic hypothesis and hospital discharge. In SIM, we identified the individuals and the date of death. We used SICLOM to identify people who were initiating ART for prospective follow-up.

In order to measure the effectiveness, the participants' records were investigated for at least 12 months after initiating ART in the following outcome variables: 1) **Virologic response**: a) Viral suppression (viral load <50copies/mm$^3$) at 6 and 12 months after initiating ART; b) time until viral suppression (time from last viral load before ART to the first that was <50copies/mm$^3$); c) virologic treatment failure when participants presented at least two consecutive tests of viral load (VL) >50copies/mm$^3$ after 6 months on ART, with an interval of less than 6 months between them, or a level of VL>1,000copies/mm$^3$.[2] 2) **Immunologic response**: a) an increase of at least 50 cells in the CD4 count in up to 12 months in comparison to when treatment was initiated[5]; b) average monthly increase in the CD4 cell count. 3) **Clinical failure**: proportion of individuals who evolved to hospitalization and death. 4) **Regimen change**: proportion of individuals who changed or modified at least one component of the ART regimen (through adverse events, virologic failure and other causes).

The reasons for modifying the ART regimen were classified as they were recorded in medical records, characterized as: adverse events from drugs, virologic failure and other reasons, including non-adherence, antiretroviral unavailable at the pharmacy, patient request, medical decision, and drug interaction. The measure used to suggest adherence to therapy was assiduity in dispensing at the pharmacy enabled by the SICLOM system, in addition to information described in the medical records during consultations. Participants were considered as assiduous when they were registered at SICLOM as having collected their monthly, bimonthly or quarterly ART according to the justification described in the system. Treatment abandonment was considered as failing to collect ART from the pharmacy within 100 days after the previously planned dispensation date.[2]

Characterization of clinical conditions when initiating ART followed the Rio de Janeiro/Caracas, BRAZIL, 1998 score. Data were collected and gathered on standardized forms by trained staff.

## Data analysis

The database was built with double entry and was later compared and corrected. We used Stata 12 for the analysis.

To analyze the virologic response, the frequency of individuals with a VL<50 copies/mm$^3$ were calculated during the first 6 months of treatment, and only those who had repeated viral load tests during this period were considered. The groups were compared using the Pearson's chi-square test. For the incidence ratio and in the person-time observation stage, the time was calculated between the ART initiation date and the date on which the first VL measurement was below 50 copies/mm$^3$. We used the Cox proportional hazard calculation to estimate the association between the exposure (antiretroviral) and the outcome (virologic response). The Kaplan Meier curve was used to analyze the time until viral suppression and for the clinical outcome related to hospitalization, considering the time of study follow-up. Individuals who changed regimens before completing 28 days of treatment were excluded. We considered this time criteria, because it was the habitual period they returned to the health service to receive their medication. Follow-up was discontinued for those who changed to ART regimens other than those in the study groups, while those who changed to one of the studied regimens were considered as new exposure and the outcomes were independently observed. Cases confirmed as lost to follow-up were censored on the date of the last follow-up recorded in the medical records.

To assess the immunologic response, participants with at least two CD4 cell count tests were considered eligible, the first count being <90 days before or up to 30 days after initiating ART. The immunologic response was observed for up to 15 months after exposure.

Participants were excluded if the first CD4 tests after initiating ART was after 12 months. To analyze the cumulative incidence and factors associated with the immunologic response (an increase in the CD4 count of at least 50 cells/mm$^3$), the multilevel generalized linear latent and mixed model (GLLAMM) was used since it is a sample with repeated measures and with varying time intervals between the tests in the period up to 15 months. We used the multilevel linear mixed-effect regression model to compare the mean increase in CD4/month by comparing antiretroviral regimens. To minimize the effect of observation dependence, the odds ratios confidence intervals and incidence of immune response were adjusted by the random effect. For the multivariate analysis, variables were included with p <0.10 to verify association with immune response.

The incidence rate, hazard ratio and *p* value were verified using the Cox proportional hazard model to investigate the sociodemographic and clinical variables as potential confounders of the virologic and immunologic responses. The variables studied were gender; sexual orientation; age group; living with a family member, friend or partner; schooling; comorbidity; clinical situation; history of hospitalization before ART; hospitalization after ART; disease after initiating ART; ART adherence failure; CD4 when initiating ART. To test the association with virologic and immunologic responses in the multivariate analyzes, those with p<0.10 were inserted into the model.

In the assessment of clinical failure, the groups were compared regarding hospitalization through survival analysis (Kaplan Mayer). For the outcome of death, the cases were described with absolute numbers and proportions amongst the users of the regimens in the different groups. Regimen changes were described in simple frequencies and compared using the Pearson's chi-square test or Fisher's exact test.

## Results

From a cohort of 840 HIV-infected patients who had initiated antiretroviral treatment, comprising a total of 898 observations, 65.1% of whom were male (N = 547). The mean follow-up period was 27.8 months, with a minimum of 28 days and a maximum of 78 months. Through differentiating the regimens, the mean follow-up time in the study was 26 months for the *tenofovir/lamivudine/efavirenz* and *tenofovir/lamivudine/lopinavir/ritonavir* regimens, 29 months for the *tenofovir/lamivudine/atazanavir/ritonavir* and 31 months for the *zidovudine/lamivudine/efavirenz*. In terms of education, 48.9% of the participants in the sample had attended school for more than 9 years and 4.2% had never attended. Most individuals (87%) came from the city of Recife or metropolitan region. With regard to sexual orientation, 55.7% of the sample identified themselves as heterosexual. The main reason recorded for HIV testing amongst participants was the presence of symptoms possibly associated with opportunistic diseases (48.5%) followed by a positive diagnosis of a partner or former partner (24.7%). The mean time from HIV diagnosis to initiating antiretroviral treatment was 21 months (± 36.4), with 19 months for participants in the prospective stage and 23 months for those in the retrospective group.

On initiating ART, 22.3% presented with a diagnosis of some other sexually transmitted disease besides HIV, of which syphilis was the most frequent (15.1%), and 23.3% presented with other diagnoses, whereby hypertension was the most common (34%).

During the follow-up period, 43 (5.1%) individuals abandoned treatment and there were 15 deaths (1.8%), with no difference between groups. The most commonly used regimens for initiating treatment were two NRTI (Nucleoside Reverse Transcriptase Inhibitors) drugs and one NNRTI (Non Nucleoside Reverse Transcriptase Inhibitors), which corresponded to the majority of the sample (88.6%). The mean age was 38 years (18 to 74 years). The age group with the largest number of participants (50%) was 25 to 39 years old. In terms of the immunologic

status on initiating ART, 30.2% (n = 254) of all participants initiated treatment with severe immunodeficiency (CD4 <200 and/or manifestations of opportunistic disease). The average initial CD4 count was 324 cells/mm$^3$, a median of 296.5 (114; 445). At the beginning of the regimen, 52% (n = 437) presented with some type of AIDS-defining symptom or disease.

## Virologic response

The overall incidence rate of virologic response (VL<50copies/mm$^3$) at 12 months was 14.61 per 100 people treated-month (95%CI:13.58–15.73). The frequency of this response was 79.6% (n = 715) in the first year of ART. Of the 451 participants who repeated a viral load test at 6 months, 366 (81.5%) presented a virologic response within the first 6 months of treatment. Amongst those who initiated ART with the tenofovir/lamivudine/efavirenz regimen, 88.4% presented a virologic response within one year, followed by the tenofovir/lamivudine/lopinavir/ritonavir regimen with 86.1%, the tenofovir/lamivudine/atazanavir/ritonavir regimen with 81.6% and the zidovudine/lamivudine/efavirenz regimen with 76%. There was a difference when comparing the proportions between the groups with NNRTIs, with a better virologic response in the group using tenofovir/lamivudine/efavirenz (p = 0.004), which did not occur amongst participants using the Protease Inhibitor (PI) based regimen.

The Fig 2 presents the survival curves that represent the time to viral suppression taking into account all observations during the cohort follow-up (2a) and between the separate analyzed groups (2b).

When comparing regimens in the group of participants who remained on the same regimen during the follow-up, there was no differences between the survival curves for virological response by regimens (Fig 3A). The same we did found for those group who changed their regimen during the follow-up. (Fig 3B). When comparing the survival curves of the virological response between these two groups of participants, we observed differences in their survival curves (p<0.001) (Fig 3C).

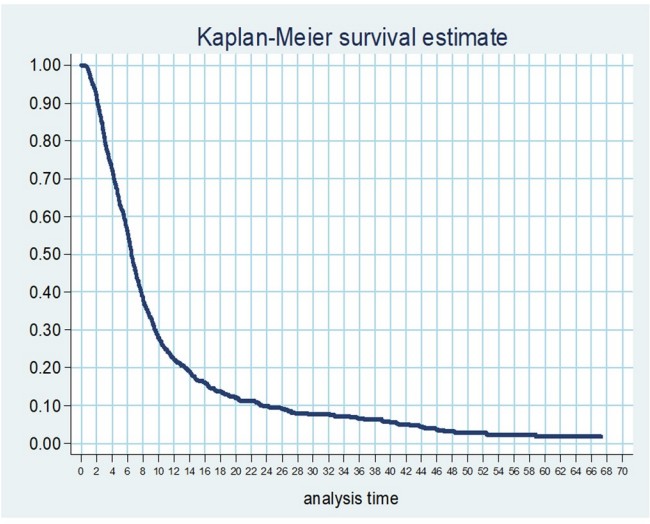

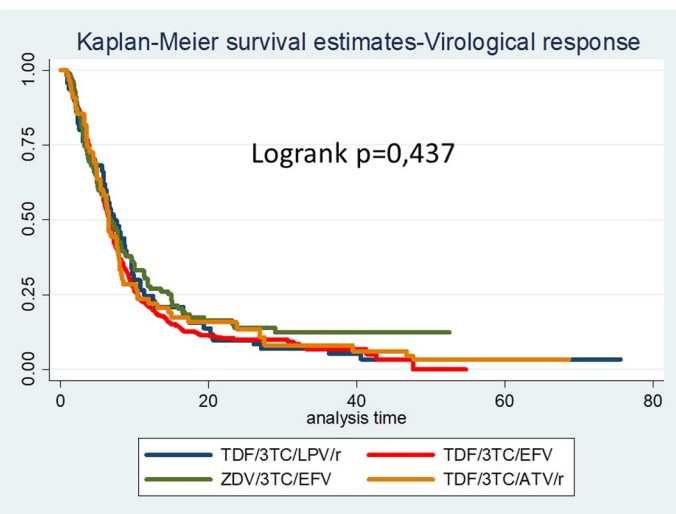

a. Time to viral supression taking into account all observations during the cohort follow-up.

b. Time to viral supression between the separete analyzed regimen groups.

**Fig 2. Survival curves for virologic response considering total observations during follow-up and different regimens.**

*a: Comparison of differences between regimens in participants who did not change treatment during follow-up.*

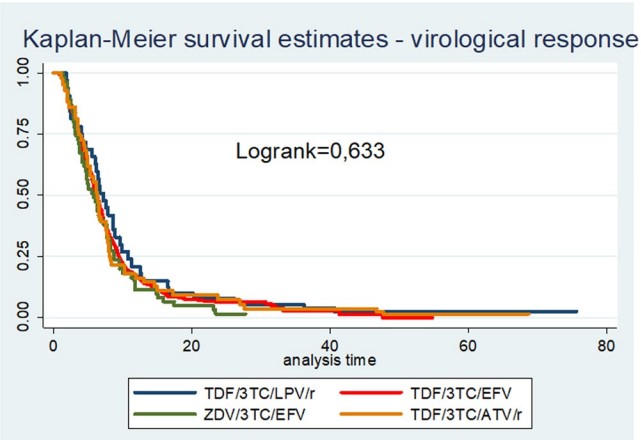

*b. Comparison of differences between regimens in participants who changed treatment.*

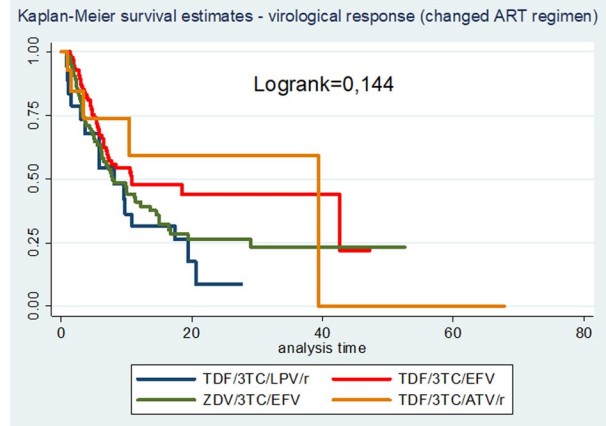

*c. Comparison of virological response survival curves between participants who changed treatment during follow-up and those who did not change treatment*

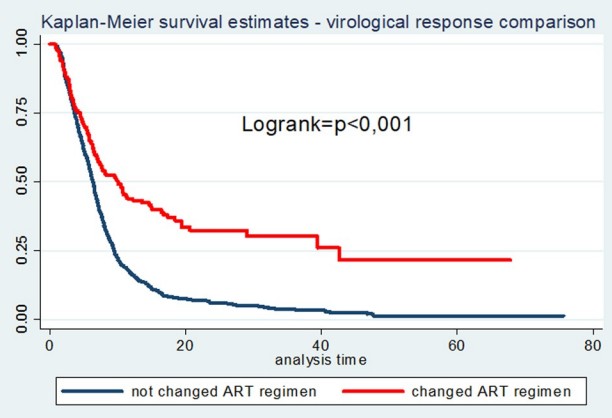

**Fig 3. Survival curves for virologic response by regimen, considering regimen change.**

The incidence rate of virologic response amongst the individuals who did not change regimen was 11.69, and amongst those who did, it was 5.33 per 100 person-months at follow-up, respectively.

The variables selected for being associated with a virologic response in the univariate analysis were: gender; sexual orientation; living with a family member/partner; comorbidity; clinical situation; hospitalization for AIDS-defining illness before ART; hospitalization for AIDS-defining illness after initiating ART; disease after initiating ART; treatment adherence failure; CD4 count when initiating ART (Table 1). The proportionality of the risk ratio was tested in all variables (Schoenfeld Residue), and the risk was proportional in all of them.

After the multivariate analysis, the characteristics associated with a virologic response were: being female; living with a family member/partner; being asymptomatic when initiating ART; not report of adherence failure and initiating ART with a CD4 count of between 200–350 cells/mm$^3$. There was no difference between the groups regarding the regimens after adjustment (Table 2).

**Table 1.** Incidence rate, HR and *p* value for the association of virological response between sociodemographic factors, clinical and immunological factors of patients who started treatment for HIV-AIDS in two referral services in Pernambuco.

| Variables | Virological Response (number) | Incidence rate (100 people / month) | HR (IC 95%) | *p*-value | *p*-value[a] |
|---|---|---|---|---|---|
| **Biological variables** | | | | | |
| **Sex** | | | | | |
| Male | 460 | 8.87 (0.08–0.10) | 1 | - | - |
| Female | 255 | 11.65 (0.10–0.13) | 1.27 (1.09–1.48) | 0.002[†] | 0.658 |
| **Sexual orientation**[*] | | | | | |
| Heterosexual | 399/614 | 8.82 (0.07–0.1) | 1 | | |
| Homo/bisexual | 215/614 | 10.39 (0.09–0.11) | 1.15 (0.98–1.36) | 0.09 | 0.588 |
| **Age group** | | | | | |
| 18–39 years old | 422 | 9.30 (0.08–0.10) | 1 | | |
| 40–59 years old | 272 | 10.25 (0.09–0.11) | 1.05 (0.90–1.22) | 0.541 | |
| 60 and over years old | 21 | 11.85(0.08–0.18) | 1.21 (0.78–1.88) | 0.385 | 0.083 |
| **Living with a family member/partner** | | | | | |
| Yes | 418/467 | 9.61 (0.08–0.10) | 1 | | |
| No | 49/467 | 6.33 (0.04–0.08) | 0.64 (0.48–0.87) | 0.004 | 0.456 |
| **Education level** | | | | | |
| 0 to 8 years of study | 188/548 | 9.36(0.08–0.11) | 1 | | |
| 9 to 11years of study | 179/548 | 10.60 (0.09–0.12) | 1.12 (0.9–1.38) | 0.273 | |
| 12 and over years of study | 181/548 | 10.85(0.09–0.12) | 1.14 (0.93–1.41) | 0.186 | 0.341 |
| **Comorbidity** | | | | | |
| Yes | 175/674 | 12.18(10.51–14.13) | 1 | | |
| No | 499/674 | 9.22(8.45–10.07) | 0.81(0.68–0.96) | 0.015[†] | 0.109 |
| **Cliical situation** | | | | | |
| Asymptomatic | 366 | 12.29(0.11–0.14) | 1 | | |
| Symptomatic | 349 | 7.94 (0.07–0.09) | 0.65 (0.56–0.75) | 0.000[†] | 0.758 |
| **Hospitalization for AIDS-defining illness before ART** | | | | | |
| Yes | 168 | 7.51(0.06–0.09) | 1 | | |
| No | 547 | 10.65(0.10–0.11) | 1.36(1.14–1.62) | 0.000[†] | 0.189 |
| **Hospitalization for AIDS-defining illness after ART** | | | | | |
| Yes | 166/714 | 7.64(0.06–0.09) | 1 | | |
| No | 548/714 | 10.55(0.10–0.11) | 1.27 (1.07–1.51) | 0.006[†] | 0.178 |
| **Disease after initiating ART** | | | | | |
| Yes | 245/714 | 8.41(0.07–0.09) | 1 | | |
| No | 469/714 | 10.53(0.10–0.11) | 1.17(1.00–1.36) | 0.048[†] | 0.107 |
| **Treatment adherence failure** | | | | | |
| Yes | 83 | 12.00 (0.09–0.14) | 1 | | |
| No | 546 | 17.26(0.16–0.18) | 1.62(1.28–2.05) | 0.000[†] | 0.534 |
| **CD4 count when initiating ART** | | | | | |
| <200 cell/mm$^3$ | 188/587 | 26.56 (0.06–0.08) | 1 | | |
| >200<350 cell/ mm$^3$ | 149/587 | 13.31 (0.09–0.13) | 1.60 (1.30–2.00) | 0.000[†] | |
| >350<500 cell/mm$^3$ | 121/587 | 10.11(0.10–0.14) | 1.58 (1.26–1.99) | 0.000[†] | |
| >500 cell/mm$^3$ | 129/587 | 11.11 (0.10–0.14) | 1.52 (1.21–1.90) | 0.000[†] | 0.255 |

[a] Risk proportionality test (schoenfeld residue)

[†] Statistically significant risk (p <0.05

**Table 2. Adjusted HR and p-value for association between virological response and sociodemographic, clinical and immunological factors of patients who started treatment for HIV-AIDS, in two referral services in Pernambuco.**

| Variables | HR (IC95%**) | *p*-value |
|---|---|---|
| **Regimen** | | |
| *tenofovir/lamivudine/lopinavir/ritonavir* | 1 | |
| *tenofovir/lamivudine/efavirenz* | 1.18(0.69–2.00) | 0.535 |
| *zidovudine/lamivudine/efavirenz* | 1.12(0.63–2.00) | 0.677 |
| *tenofovir/lamivudine/atazanavir/ritonavir* | 0.90(0.45–1.78) | 0.763 |
| **Sex** | | |
| Female | 1.29(1.03–1.60) | 0.024 |
| **Living with a family member/partner** | | |
| Yes | 1.69(1.18–2.42) | 0.004 |
| **Clinical situation** | | |
| Asymptomatic | 1.37(1.05–1.77) | 0.019 |
| **Treatment adherence failure** | | |
| No | 3.32(2.42–4.54) | 0.000 |
| **CD4 count when initiating ART** | | |
| **>200<350 cell/ mm$^3$** | 1.69(1.24–2.30) | 0.001 |
| **>350<500 cell/mm$^3$** | 1.22(0.87–1.71) | 0.245 |
| **>500 cell/mm$^3$** | 1.33(0.94–1.88) | 0.101 |

** Adjusted HR (Cox Test Multivariate Analysis)

## Immunologic response

To analyze the immunologic response, the sample consisted of 535 participants, out of a total of 555 observations. Of the total participants, 82.5% presented immunologic response (an increase of at least 50 cells/mm$^3$ of CD4 T lymphocytes) within up to 15 months of treatment. Most were male (62.8%), with a mean age of 39.2 years (± 10.6), who had attended school for more than 9 years (65.7%), and 23.4% presented with comorbidities and other sexually transmitted infections. More than half of the participants (52.9%) presented some symptoms suggestive of AIDS on initiating ART. The median CD4 count before initiating ART was 281.5 (104; 420.5) with an average of 305.2 cells. Most individuals (57.6%) initiated ART with an initial CD4 cell count below 350, 18.3% were between 350 and 500 and 20.7% with a CD4> 500. At the end of follow-up, 227 participants (42.4%) presented CD4 counts >500. Treatment adherence failure was recorded in 17.8% of the participants.

When the group of participants was compared with the 305 losses, both had similar characteristics, most were male, with a mean age of 39.2 ± 10.6 years for the analyzed group and 31.5 years for the group considered as lost and both had more than 60% of individuals with more than 9 years of schooling.

In the univariate analysis, the chance of obtaining an immune response was higher in younger individuals, with a CD4 count on initiating ART of below 500 cells/mm3 and in individuals with no record of treatment adherence failure within 1 year (Table 3).

The immunologic response was higher than 75% in all regimen groups studied, with no observed difference between them (Table 4).

For the multivariate analysis, the following variables were included in the model: age group, adherence failure and initial CD4 cell count, to verify association with immune response.

Treatment adherence failure was associated with an inadequate immunologic response and initiating ART with a CD4 count above 500 cells/mm$^3$ continued as a factor associated with a

**Table 3. Raw OR and p value for the association between immune response and sociodemographic factors, clinical and immunological factors of patients who started antiretroviral treatment at two referral services in Pernambuco.**

| Variables | OR (IC 95%)* | p-value* | OR(IC 95%)** | p-value** |
|---|---|---|---|---|
| **Sex** | | | | |
| Male | 1 | | | |
| Female | 0.77(0.49–1.23) | 0.276 | - | - |
| **Age Group** | | | | |
| 18–39 years old | 2.62 (0.83–8.29) | 0.045 | 2.99(0.99–11.29) | 0.031[†] |
| 40–59 years old | 3.40 (1.02–11.13) | 0.101 | 183(0.91–2.47) | 0.033[†] |
| >60 years old | 1 | | 1 | |
| **Living with a family member/partner** | | | | |
| Yes | 1 | | | |
| No | 1.08 (0.46–2.52) | 0.866 | - | - |
| **Education level** | | | | |
| 0 to 8 years of study | 1 | | | |
| 9 to 11years of study | 1.05 (0.57–1.93) | 0.864 | - | - |
| 12 and over years of study | 0.93 (0.52–1.69) | 0.823 | - | - |
| **Comorbidity** | | | | |
| Yes | 1 | | | |
| No | 1.43 (0.82–2.48) | 0.203 | - | - |
| **Other sexually transmitted infections** | | | | |
| Yes | 1 | | | |
| No | 1.18 (0.70–2.00) | 0.518 | - | - |
| **Clinical situation** | | | | |
| Asymptomatic | 1 | | | |
| Symptomatic | 1.48 (0.92–2.37) | 0.113 | - | - |
| **Treatment adherence failure** | | | | |
| Yes | 1 | | 1 | |
| No | 2.30 (1.17–4.53) | 0.015 | 2.70(1.26–5.78) | 0.010[†] |
| **CD4 count when initiating ART** | | | | |
| <200 cell/mm$^3$ | 3.83 (2.11–6.94) | <0.001 | 4.23(1.91–9.37) | <0.001[†] |
| >200<350cell/ mm$^3$ | 2.96 (1.52–5.74) | <0.001 | 3.38(1.66–6.91) | <0.001[†] |
| >350<500 cell/mm$^3$ | 2.00 (1.05–3.79) | 0.034 | 2.29(1.17–4.50) | 0.015[†] |
| >500 cell/mm$^3$ | 1 | | 1 | |

[†] Statistically significant risk ($p < 0.05$)

*Gross odds ratio

** Adjusted OR (GLLAMM Multilevel Model)

lower frequency of response. We observed no statistical difference between the analyzed regimens (Table 4).

The multilevel mixed-effect linear regression model demonstrated that the mean increase in CD4 cells per month compared to the antiretroviral regimens was similar within all the groups (Table 5).

## Clinical failure

The incidence proportion of hospitalization was 7.79 per 100 people (6.79–8.94). Thehe incidence of hospitalization among those with CD4 < 200 was significantly higher compared to those with Cd4 count > = 200 (p< 0.001).

**Table 4. Comparison of immunological response between antiretroviral regimens, considering CD4 cell increase of 50 or more over 1 year, in two reference services in Pernambuco.**

| Esquema | No. comments | No. Imune response | % (95% CI)* | OR (IC 95%)** | *p*-value |
|---|---|---|---|---|---|
| *tenofovir/lamivudine/ efavirenz* **(1)** | 360 | 281 | 78(73.7–82.4) | 1.11(0.43–2.84) | 0.833 |
| *zidovudine/lamivudine /efavirenz* **(2)** | 111 | 90 | 81 (73.7–88.5) | 1.33(0.47–3.80) | 0.592 |
| *tenofovir/lamivudine/ lopinavir/ritonavir* **(3)** | 41 | 31 | 75.6 (61.9–8.9) | 1 | |
| *tenofovir/lamivudine/ atazanavir/ritonavir* **(4)** | 43 | 33 | 76.7 (63.6–89.9) | 1.67(0.48–5.80) | 0.899 |

*Cumulative incidence

**Adjusted OR- Multilevel Model (GLLAMM)

***OR between regimens (1 versus 2; 3 versus 4)

With regard to hospitalization, there was no difference between the groups using PI-based regimens (16.8%) when compared to those using NNRTIs (15.4%). When comparing those using NNRTI regimens, the group using zidovudine/lamivudine/ efavirenz presented a higher frequency of hospitalizations at 1 year (21.3%) with a significant difference compared to the tenofovir/lamivudine/efavirenz regimen (13, 9%), with p = 0.032. There was no difference amongst those using PIs (p = 0.317).

Of the 15 deaths that occurred during the cohort follow-up, 11 were male. The proportion of users from the tenofovir/lamivudine/lopinavir/ritonavir regimen was 4.3% (2/47), followed by the tenofovir/lamivudine/atazanavir/ritonavir regimen with 1.7% (1/57), the tenofovir/lamivudine/efavirenz with 1.7% (10/518) and the zidovudinar/lamivudine/efavirenz with 1.3% (2/158). The median initial CD4 cell count on initiating ART in cases of death was 202 (55; 229). The main causes involved opportunistic diseases such as AIDS complications (11/15), followed by lung, colon and liver cancer (4/15).

## Regimen changes

Regimens were changed in 28.1% (0.25–0.31) of the observations (n = 254) and 63.8% (n = 162) of the changes were due to virologic failure and adverse events. The regimen with the lowest rate of change for any cause was tenofovir/lamivudine/efavirenz (16.7%). When compared at 1 year of ART treatment, the regimen with the highest proportion of change for any reason was the zidovudine/lamivudine/efavirenz when compared to the tenofovir/lamivudine/efavirenz (p<0.001). Adverse events were the reason for change in 46.5% (0.40–0.53), and were the main reason for regimen changes in the cohort. Amongst the groups, considering the first 6 months of treatment, that group taking zidovudine/lamivudine/efavirenz changed regimen to a greater extent by adverse events when compared to tenofovir/lamivudine/efavirenz (p = 0.04). There was no difference between IP users.

**Table 5. Mean increase in CD4 T lymphocytes / month comparing by ARV regimen.**

| Regimen | Average increase No. cells / month | % (IC 95%)* |
|---|---|---|
| *tenofovir/lamivudine/ efavirenz* **(1)** | 6.7 | (6.41–7.52) |
| *zidovudine/lamivudine /efavirenz* **(2)** | 6.5 | (5.98–7.06) |
| *tenofovir/lamivudine/ lopinavir/ritonavir* **(3)** | 7.8 | (6.75–8.87) |
| *tenofovir/lamivudine/ atazanavir/ritonavir* **(4)** | 5.33 | (3.95–6.71) |

* Multilevel mixed effect linear regression model

Of the participants who changed regimens, 17.3% (0.13–0.22) did so because of virologic failure (n = 44). The mean time between detecting virologic failure and changing the regimen was 4.8 months. The least changed regimen was tenofovir/lamivudine/efavirenz (16.7%). In the first year of ART, there was no regimen change because of virologic failure in the groups using protease inhibitors, and no difference was observed between the groups that changed due to virologic failure. Participants with a pre-treatment CD4<200 changed the initial regimen more frequently (37.2%) than those with a CD4>200 (23.4%) (p<0.001).

## Discussion

We compared the effectiveness of four ART regimens at two referral hospital services in the state of Pernambuco/Brazil in a bidirectional cohort. The proportion of participants with a virologic response within the first 6 months of taking ART was higher in the tenofovir/lamivudine/efavirenz group when compared to the zidovudine/lamivudine/efavirenz group, but did not differ from the others. There was no difference between the regimens in relation to the time until viral suppression and virologic failure. There was no difference between the groups regarding immune response and the mean monthly increase in the CD4 count. Regimen changes for any reason was more frequent in the zidovudine/lamivudine/efavirenz group. The proportion of hospitalizations was also higher in the zidovudine/lamivudine/efavirenz group when compared to the tenofovir/lamivudine/efavirenz group. In general, adverse events were the main reason for changing regimens. Amongst the NNRTI users, the zidovudine/ lamivudine/efavirenz group presented a higher chance of changing regimens for this reason. Participants who did not change the ART regimen during follow-up demonstrated a better virologic response, as did females, those living with partners or relatives, asymptomatic individuals initiating ART, and those who initiated treatment with a CD4>200. A better immune response outcome was observed in younger people, those with no adherence failure, and CD4<500. Our sample is similar to other cohorts with PLWHIV regarding general sociodemographic characteristics, whereby the cases were mostly concentrated in heterosexual males with more than 9 years of schooling [9, 10].

In order to assess the effectiveness of antiretroviral regimens, virologic response has been used as the most practical and earliest outcome [4]. In the present study, a virologic response was observed in almost 80% of participants at 1 year, which is similar to reports in other cohorts and systematic reviews [10–12]. Amongst the participants who repeated the viral load test at 6 months, more than 80% demonstrated a virologic response. No differences were observed in the virologic response between the analyzed regimens regarding VL suppression at 1 year. However, during the 6-month period, the tenofovir/lamivudine/efavirenz group presented a higher response rate when compared to the zidovudine/lamivudine/efavirenz. This fact may be explained by a change from the tenofovir/lamivudine/efavirenz regimen to a fixed-dose combined presentation, which may have favored participant adherence [12, 13]. A review study analyzing randomized controlled trials and observational studies reported no difference between tenofovir and zidovudine in relation to virologic response, but observed higher rates of adherence and immune response in regimens containing tenofovir [12]. In a meta-analysis comparing outcomes in clinical trials with zidovudine/lamivudine versus abacavir/lamivudine versus tenofovir/emtricitabine regimens in fixed-dose combinations at 12 and 24 months, the authors reported a more efficient virologic response to tenofovir/emtricitabine in comparison to others, with the third drug composed of a PI or dolutegravir [11]. Dadi, Kefali and Mega, et al [14] reported greater virologic suppression and tolerability in the tenofovir/emtricitabine/efavirenz group in comparison to the zidovudine/lamivudine/efavirenz group as initial ART for treatment naïve participants.

In the multivariate analysis of our data, females presented a higher chance of virologic response, thereby corroborating an observational cohort conducted in Uganda from 2004 to 2011 with 4537 participants [15]. Adherence may explain these differences [16].

Living with a partner or family member was associated with a better virologic response, probably because social bonding has a strong influence over not feeling lonely or depressed, and adhering and responding better to medication [17]. Onoya et al. in South Africa [18] observed that the risk of virologic failure was up to 5 times higher in non-adherence cases. Lee, et al, 2017 [19] reported that virologic failure after 6 months of ART is associated with a substantial risk of death. There is a drop of 14% in the risk of death for individuals with rapid viral suppression.

In terms of the immunologic response, we observed no difference between the groups in the first year of treatment, nor regarding the mean monthly increase in the CD4 count. When adjusted for confounders, an increase in the CD4 was higher in younger people, thereby corroborating other studies [20, 21]. In addition, individuals with no record of therapy adherence failure were 2.3 times more likely to have an immunologic response. It is commonly agreed that taking ART correctly is important to control the viral load and consequently restore immunity [16]. Failure to adhere seems to be the main factor associated with poorer outcomes [22, 23].

Participants with initial CD4<200 had a higher chance of immunologic response with the criterion of a minimum increase of 50 cells after 6 months of treatment. However, it is known that individuals with a severe immunologic condition may either take longer or indeed never manage to reach a CD4 immune recovery threshold at normal levels, even maintaining virologic suppression [19]. It is possible that with a lower pretreatment CD4 count, there may be a temporary increase in CD4, but without restoring it to normal levels (CD4>500) [24]. In a 3-year cohort of 1327 participants, it was observed that 75% of those who initiated ART with a CD4<200 did not achieve complete immune recovery [25]. Other authors have argued that regardless of the CD4 count, the time spent with HIV until receiving treatment is the major factor influencing CD4 recovery [19, 26].

In our study, initiating ART with a lower CD4 count was more often associated with changes in the initial regimen and with a greater need for hospitalization. Initiating ART in the first 6 months after infection is considered early [4, 27]. A study by the ART-CC [28] group also observed that lower CD4 counts before initiating antiretroviral treatment is a risk factor for regimen changes [29].

Several authors state that the pretreatment CD4 count acts as a prognostic indicator for the effectiveness outcome [20, 21, 24, 29]. Robbins et al [30] observed almost twice as much propensity for therapy failure in participants who initiated ART with a CD4 <200 when compared to those who initiated with higher CD4 counts.

The group with the highest ART regimen change rate for any reason during follow-up was the group on zidovudine/lamivudine/efavirenz. A significant part of these changes (31.3%) was to simplify the regimen. During the follow-up period, the Ministry of Health advocated changing this regimen to tenofovir/lamivudine/efavirenz in a fixed-dose combination in individuals with controlled viremia.

In this cohort, adverse events were the main cause for changing the regimen, which is often the most common reason for regimen change [31], and is generally most evident in the first trimester of treatment. The proportion of adverse events in the group using zidovudine/lamivudine/efavirenz group was higher in the NNRTI regimens users. Zidovudine may cause bone marrow supression, with severe anemia, which is one of the reasons to change and for being hospitalized for blood transfusion [32].

There was no difference between the regimens with regard to change because of failure. In our findings, during the first year, there were no changes due to failure amongst the groups

using PI-based regimens. This reinforces the safety of these drugs in terms of potency [33, 34]. For participants who changed regimens during follow-up, we observed an association with a lower virologic response, suggesting that at least some of them changed regimens because a detectable viral load had been maintained for more than one year. Consultations undertaken at SISGENO served as an instrument to confirm virologic failure in cases where genotyping was requested (3.7% of the sample). Virologic failure was observed after 6 months of treatment and was the second leading cause of antiretroviral regimen change in both this and other studies [29, 31].

In terms of clinical failure, when comparing the groups, we observed no differences in the proportion of hospitalizations at 1 year amongst PI-based regimen users. A higher frequency of hospitalizations was observed amongst the NNRTI users, in the zidovudine/lamivudine/efavirenz group, when compared to the tenofovir/lamivudine/efavirenz group.

Investigating effectiveness in a clinical cohort portrays the "real" situation of the exposed population and the inclusion criteria are less "restrictive" than in randomized trials, thereby allowing for different information on treatment response in populations that may not be well represented in clinical trials. One possible limitation of this study was collecting information from secondary sources (medical records). In order to minimize information gaps and losses, we complemented data collection with national information systems on laboratory data, drug supplies, hospitalizations and mortality. The field team used a single instrument, standardized after pilot testing and training for data capture. All inpatient data were collected from the Hospitalization Information System, and no access was made to private services or generated through other instances, but Hospitalization in private services must have been an exception in our cohort. To minimize the possibility of underestimating hospitalization data, the hospital records of participants who were not registered in the SIH were reviewed.

In this study we observed few differences between the compared regimens with regard to the main outcomes. Virologic response in the sample was achieved in almost 80% within the first year of treatment. All the regimens seemed to be effective in the comparison groups. Differences in outcomes may be due more to individual characteristics of the participant, toxicity of certain medications, acceptability, and dosage rather than to drug potency. In order to identify the most appropriate regimens for initiating ART, several factors, including treatment costs, need to be taken into consideration, and we suggest further studies to compare regimens with similar effectiveness, their respective costs and financial impacts on the Brazilian Integrated Healthcare System.

## Acknowledgments

Correia Picanço Hospital and Oswaldo Cruz University Hospital for friendly suporting access to outpatient and inpatient medical records research.

## Author Contributions

**Conceptualization:** Aracele Tenório de Almeida e Cavalcanti, Demócrito de Barros Miranda-Filho.

**Data curation:** Ricardo Arraes de Alencar Ximenes, Ulysses Ramos Montarroyos, Demócrito de Barros Miranda-Filho.

**Formal analysis:** Aracele Tenório de Almeida e Cavalcanti, Ricardo Arraes de Alencar Ximenes, Demócrito de Barros Miranda-Filho.

**Funding acquisition:** Demócrito de Barros Miranda-Filho.

**Investigation:** Aracele Tenório de Almeida e Cavalcanti, Polyana Monteiro d'Albuquerque, Rosário Antunes Fonseca, Demócrito de Barros Miranda-Filho.

**Methodology:** Aracele Tenório de Almeida e Cavalcanti, Ricardo Arraes de Alencar Ximenes, Demócrito de Barros Miranda-Filho.

**Project administration:** Aracele Tenório de Almeida e Cavalcanti, Demócrito de Barros Miranda-Filho.

**Resources:** Aracele Tenório de Almeida e Cavalcanti, Polyana Monteiro d'Albuquerque, Rosário Antunes Fonseca.

**Software:** Ulysses Ramos Montarroyos.

**Supervision:** Aracele Tenório de Almeida e Cavalcanti, Demócrito de Barros Miranda-Filho.

**Validation:** Aracele Tenório de Almeida e Cavalcanti, Ricardo Arraes de Alencar Ximenes, Ulysses Ramos Montarroyos.

**Visualization:** Demócrito de Barros Miranda-Filho.

**Writing – original draft:** Aracele Tenório de Almeida e Cavalcanti.

**Writing – review & editing:** Aracele Tenório de Almeida e Cavalcanti, Demócrito de Barros Miranda-Filho.

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
