## [Decision Letter · Decision Letter 0]

9 Mar 2020

PONE-D-20-00907

Effectiveness analysis of virological response, immune response, regimen change, and clinical failure of four different antiretroviral regimens for treating people living with HIV: an observational cohort

PLOS ONE

Dear Mrs Cavalcanti,

Thank you for submitting your manuscript to PLOS ONE. After careful consideration, we feel that it has merit but does not fully meet PLOS ONE’s publication criteria as it currently stands. Therefore, we invite you to submit a revised version of the manuscript that addresses the points raised during the review process.

We would appreciate receiving your revised manuscript by Apr 23 2020 11:59PM. To enhance the reproducibility of your results, we recommend that if applicable you deposit your laboratory protocols in protocols.io, where a protocol can be assigned its own identifier (DOI) such that it can be cited independently in the future. For instructions see: http://journals.plos.org/plosone/s/submission-guidelines#loc-laboratory-protocols

We look forward to receiving your revised manuscript.

Kind regards,

Marcel Yotebieng, M.D., MPH, Ph.D

Academic Editor

PLOS ONE

Journal Requirements:

1. Please ensure that your manuscript meets PLOS ONE's style requirements, including those for file naming. The PLOS ONE style templates can be found at http://www.plosone.org/attachments/PLOSOne_formatting_sample_main_body.pdf and http://www.plosone.org/attachments/PLOSOne_formatting_sample_title_authors_affiliations.pd

2. We note that you have reported significance probabilities of 0 in places. Since p=0 is not strictly possible, please correct this to a more appropriate limit, eg 'p<0.0001'.

Reviewers' comments:

Reviewer's Responses to Questions

**Comments to the Author**

1. Is the manuscript technically sound, and do the data support the conclusions?

Reviewer #1: Yes

Reviewer #2: Yes

2. Has the statistical analysis been performed appropriately and rigorously? 

Reviewer #1: Yes

Reviewer #2: Yes

3. Have the authors made all data underlying the findings in their manuscript fully available?

Reviewer #1: Yes

Reviewer #2: Yes

4. Is the manuscript presented in an intelligible fashion and written in standard English?

Reviewer #1: Yes

Reviewer #2: No

5. Review Comments to the Author

Reviewer #1: This is a very comprehensive and interesting manuscript.

I have few minor comments

Why the authors decided to sample from the retrospective cohort, and not use all the available data?

It is not clear if the women who changed ART in the first year due to pregnancy are accounted in the "losses". "Women were excluded who had modified", revise language

It is not clear what the 898 observations are as opposed to the number of patients (840)

Figure 3 a and b: give more details in the text about the difference

Reviewer #2: This paper has very useful and important information. Authors explored multiple outcomes from observation cohort study among people living with HIV. Each section of the manuscript has been explored rigorously. The only concern is the English. The manuscript needs language revision.

General comment:

The paper will benefit language revision specifically in the way results were presented.

Check for consistencies of decimals in the tables

Avoid some medical terminologies where possible

Check for type errors

6. PLOS authors have the option to publish the peer review history of their article (what does this mean?). If published, this will include your full peer review and any attached files.

Reviewer #1: Yes: Barbara Castelnuovo

Reviewer #2: No

---

## [Author Response · Author response to Decision Letter 0]

30 Jul 2020

Response to Reviewers

Reviewer #1: This is a very comprehensive and interesting manuscript.

Why the authors decided to sample from the retrospective cohort, and not use all the available data?

The retrospective component is justified by the possibility of a longer follow-up of patients, allowing the study of outcomes that occur less early.

It is not clear if the women who changed ART in the first year due to pregnancy are accounted in the "losses". "Women were excluded who had modified", revise language.

Revised

It is not clear what the 898 observations are as opposed to the number of patients (840).

Second paragraph of Method- data analysis:

“ Follow-up was discontinued for those who changed to ART regimens other than those in the study groups, while those who changed to one of the studied regimens were considered as new exposure and the outcomes were independently observed.”

Figure 3 a and b: give more details in the text about the difference.

When comparing regimens in the group of participants who remained on the same regimen during the follow-up, there was no differences between the survival curves for virological response by regimens (Fig 3.a). The same we did found for those group who changed their regimen during the follow-up. (Fig 3.b). When comparing the survival curves of the virological response between these two groups of participants, we observed differences in their survival curves (p<0.001) (Fig 3.c).

Reviewer #2: This paper has very useful and important information. Authors explored multiple outcomes from observation cohort study among people living with HIV. Each section of the manuscript has been explored rigorously. The only concern is the English. The manuscript needs language revision.

Revised

The paper will benefit language revision specifically in the way results were presented.

Revised

Check for consistencies of decimals in the tables

Revised

Avoid some medical terminologies where possible

Revised

Check for type errors

Revised

Authors stated that “Individuals who changed regimens before completing 28 days of treatment were excluded” What was the criteria for the 28 days?

We considered this time criteria, because it was the habitual period they returned to the health service to receive their medication.

Virological response

“When comparing participants who changed regimens during follow-up with

those who remained on the same regimen, we observed different survival curves (Fig 3,

3.a and 3.b). There was no difference in the virologic response between the groups.

There was a difference when we compared the participants who changed the regimen

during follow-up period with those who did not (p <0.001) (Fig 3.c).”

This paragraph makes it difficult to follow: The way it is written it is not exactly what these graphs were demonstrating.

3a: compared differences of regimens among those who did not change treatment

3b. compared differences of regimens among those who changed treatment

3c. Compared survival between those who changed and those who did not change treatment. This section needs to be re-written

Revised

Tables: 

(IC 95%), can this be written as 95% CI, this is standard

Revised

Check for typo error (table 1 Cliical situation 

Revised

Check for the consistency of decimal places table 3.

3.4 (1.02-11.13) to 3.40, since you are doing 2 decimals through out.

Revised

For this paragrapgh “After the multivariate analysis, the characteristics associated with a virologic response were: being female; living with a family member/partner; being asymptomatic when initiating ART; not report of adherence failure and initiating ART with a CD4 mm3 count of between 200-350 cells/”

Revised

Would it be better to express those factors that were associated with increased or decreased likeli-hood of viral response separately?

We chose to present the multivariate analysis separately to facilitate the exposure of the data in more detail.

Legend for table 3, authors reported gross odd ratio, it would be better to just state an adjusted instead of gross

Revised

Below information is also found in the results section. This information should be in the analysis section

“For the multivariate analysis, the following variables were included in the

model: age group, adherence failure and initial CD4 cell count, with p <0.10 in the

multivariate analysis to verify association with immune response”

Revised

Clinical failure.

Incidence proportion should not be incidence rate

Revised

Below information can be better presented

“Amongst the participants who initiated ART with a CD4 count <200, 29.9% were

hospitalized during follow-up, whereas in those with higher counts, the proportion was 18.4%, with a significant difference for 1 year”

This can be written as if one says “the incidence of hospitalization among those with CD4 < 200 c/ml was significantly higher compared to those with Cd4 count >= 200. (p< 0.001)”. something like that (13, 9%) should be 13.9%

Revised

Discussion

Terminologies such as “myelotoxicity” Would avoid medical terminology and make it easier bone marrow suppression

Revised

Figures:

Needs to be improved. Groups compared are hardly visualized, poor quality. Can customize analytical codes and produce high quality figures

Revised

---

## [Decision Letter · Decision Letter 1]

9 Sep 2020

Effectiveness of four antiretroviral regimens for treating people living with HIV

PONE-D-20-00907R1

Dear Dr. Cavalcanti,

We’re pleased to inform you that your manuscript has been judged scientifically suitable for publication and will be formally accepted for publication once it meets all outstanding technical requirements.

Kind regards,

Marcel Yotebieng, M.D., MPH, Ph.D

Academic Editor

PLOS ONE

Additional Editor Comments (optional):

Reviewers' comments:

Reviewer's Responses to Questions

**Comments to the Author**

1. If the authors have adequately addressed your comments raised in a previous round of review and you feel that this manuscript is now acceptable for publication, you may indicate that here to bypass the “Comments to the Author” section, enter your conflict of interest statement in the “Confidential to Editor” section, and submit your "Accept" recommendation.

Reviewer #1: All comments have been addressed

2. Is the manuscript technically sound, and do the data support the conclusions?

Reviewer #1: Yes

3. Has the statistical analysis been performed appropriately and rigorously? 

Reviewer #1: Yes

4. Have the authors made all data underlying the findings in their manuscript fully available?

Reviewer #1: Yes

5. Is the manuscript presented in an intelligible fashion and written in standard English?

Reviewer #1: Yes

6. Review Comments to the Author

Reviewer #1: (No Response)

7. PLOS authors have the option to publish the peer review history of their article (what does this mean?). If published, this will include your full peer review and any attached files.

Reviewer #1: No

---

## [Editor Report · Acceptance letter]

16 Sep 2020

PONE-D-20-00907R1 

Effectiveness of four antiretroviral regimens for treating people living with HIV 

Dear Dr. Cavalcanti:

I'm pleased to inform you that your manuscript has been deemed suitable for publication in PLOS ONE. Congratulations! Your manuscript is now with our production department. 

Kind regards, 

on behalf of

Dr. Marcel Yotebieng 

Academic Editor

PLOS ONE